# PROSAC: Provably Safe Certification for Machine Learning Models under Adversarial Attacks

## Abstract

It is widely known that state-of-the-art machine learning models — including vision and language models — can be seriously compromised by adversarial perturbations, so it is also increasingly relevant to develop capability to certify their performance in the presence of the most effective adversarial attacks. Our paper offers a new approach to certify the performance of machine learning models in the presence of adversarial attacks, with population level risk guarantees. In particular, given a specific attack, we introduce the notion of a $(\alpha, \zeta)$ machine learning model safety guarantee: this guarantee, which is supported by a testing procedure based on the availability of a calibration set, entails one will only declare that a machine learning model adversarial (population) risk is less than $\alpha$ (i.e. the model is safe) given that the model adversarial (population) risk is higher than $\alpha$ (i.e. the model is in fact unsafe), with probability less than $\zeta$. We also propose Bayesian optimization algorithms to determine very efficiently whether or not a machine learning model is $(\alpha, \zeta)$-safe in the presence of an adversarial attack, along with their statistical guarantees. We apply our framework to a range of machine learning models — including various sizes of vision Transformer (ViT) and ResNet models — impaired by a variety of adversarial attacks such as AutoAttack, SquareAttack and natural evolution strategy attack, in order to illustrate the merit of our approach. Of particular relevance, we show that ViT's are generally more robust to adversarial attacks than ResNets and ViT-large is more robust than smaller models. Overall, our approach goes beyond existing empirical adversarial risk based certification guarantees, paving the way to more effective AI regulation based on rigorous (and provable) performance guarantees.

## 1 Introduction

With the development of increasingly capable autonomous machine learning systems and their use in a range of domains from healthcare to banking and finance, education, and e-commerce, to name just a few, policy makers across the world are in the process of formulating detailed regulatory requirements that will apply to developers and operators of AI systems. The EU is at the forefront of the drive to regulate AI systems. Proposals for an EU AI Act, an AI Liability Directive, and an extension of the EU Product Liability Directive to AI systems and AI-enabled goods are at advanced stages of the legislative process. Other jurisdictions, too, pursue a variety of regulatory initiatives. In some countries, such as the United States and the UK, these initiatives consist so far mostly in high-level principles designed to guide regulators in the interpretation and application of sector-specific regulation to AI. In others, such as China, policy makers have adopted highly detailed regulations that are often tailored to specific techniques, for example generative AI (Sheehan, 2023).

Where detailed regulation exists or has been proposed, as in the EU, it typically operates from two angles. Some regulatory instruments establish ex ante and ongoing requirements that are a precondition for the (continued) operation of an AI system. The proposed EU AI Act is a prime example of this approach. Depending on the risk level of a system, it requires, for example, an assessment of conformity with applicable standards, as well as compliance with risk management, testing, data governance, transparency, and cybersecurity requirements. Other regulatory instruments, such as

the proposed EU AI Liability Directive, seek to facilitate the recovery of damages if end users are injured as a result of the operation of an AI system.

In both cases, regulation presupposes that it is technically possible to develop certification procedures that can provide rigorous (provable) performance guarantees. The proposed AI Act requires systems classified as "high risk" to have risk management systems capable of estimating and evaluating both "the risks that may emerge when the high-risk AI system is used in accordance with its intended purpose and under conditions of reasonably foreseeable misuse" (Art. 9). The Act further stipulates that high risk systems must "achieve an appropriate level of accuracy, robustness and cybersecurity", including where attempts are made by unauthorised third parties to alter the performance of the system, i.e. where adversarial attacks occur (Art. 15(1), (4)). Levels of accuracy and robustness must be measured and disclosed to users (Art. 15(2)). The AI Liability Directive sets out rules for damages claims in the case of fault on the part of the developer of an AI system. While the Directive does not provide a harmonised definition of fault, using state-of-the-art certification procedures and disclosing performance guarantees to operators and end users will typically play a key role in evaluating legal concepts like fault and negligence under national laws that govern this question.

Developing certification procedures is not trivial due to the fact that state-of-the-art machine learning models are black-boxes that are poorly understood; furthermore, the standard train/validate/test paradigm often lacks rigorous statistical guarantees, so it is a poor certification instrument. However, recent years have witnessed the introduction of various (promising) procedures, building on recent advances in statistics, that can be used to endow black-box / complex state-of-the-art machine learning models with statistical guarantees (Bates et al., 2021; Angelopoulos et al., 2021; Laufer-Goldshtein et al., 2023). For example, (Bates et al., 2021) have proposed a framework to offer rigorous distribution-free error control of machine learning models for a variety of tasks. (Angelopoulos et al., 2021) have proposed a procedure (the Learn-then-Test framework), leveraging multiple hypothesis testing techniques, to calibrate machine learning models so that their predictions satisfy explicit, finite-sample statistical guarantees. Building upon the Learn-then-Test framework, (Laufer-Goldshtein et al., 2023) introduce a procedure to identify machine learning model risk-controlling configurations that also satisfy a variety of other objectives. Conformal prediction techniques have also been proposed to quantify the reliability of the predictions of machine learning models, e.g. (Angelopoulos & Bates, 2023).

Our paper builds on this line of research to offer an approach – PROSAC – to certify the robustness of a machine learning model under adversarial attacks (Bruna et al., 2014; Chakraborty et al., 2018). In particular, we also build upon hypothesis testing techniques akin to those in (Angelopoulos et al., 2021; Laufer-Goldshtein et al., 2023) to determine whether or not a model is robust against a specific adversarial attack. However, our approach differs from those in (Angelopoulos et al., 2021; Laufer-Goldshtein et al., 2023) because we aim to guarantee the machine learning model is safe for any attacker hyper-parameter configuration, rather than identify the machine learning model is safe for at least one such hyper-parameter configuration. PROSAC is then used to benchmark a wide variety of state-of-the-art machine learning models, such as vision Transformers (ViT) and ResNet models, against a number of adversarial attacks, such as AutoAttack, SquareAttack and natural evolution strategy attack, in vision tasks.

**Contributions**: Our main contributions are as follows:

- We propose PROSAC, a new framework to certify whether or not a machine learning model is robust against a specific adversarial attack. Specifically, we propose an hypothesis testing procedure underlying a notion of $(\alpha, \zeta)$ machine learning model safety, entailing (loosely) that the model adversarial risk is less than a (pre-specified) threshold $\alpha$ with a (pre-specified) probability higher than $\zeta$.

- We propose a Bayesian optimization algorithm — concretely, the (Improved) GP UCB algorithm — to approximate the p-values associated with the underlying hypotheses testing problems, with a number of queries that scales much slower than the number of hyper-parameter configurations available to the attacker.

- We also demonstrate that – under a slightly more stringent testing procedure – the proposed Bayesian optimization algorithm allows to rigorously certify $(\alpha, \zeta)$ safety of a specific machine learning model in the presence of a specific adversarial attack.

- Finally, we offer a series of experiments elaborating about $(\alpha, \zeta)$ safety of different machine learning models in the presence of different adversarial attacks. Notably, our framework reveals that ViTs are more robust to adversarial perturbations than ResNets, and that ViT-large is more robust than smaller models.

**Organization**: Our paper is organized as follows: The following section briefly overviews related work. Section 3 presents the problem statement, including the notion of $(\alpha, \zeta)$ machine learning model safety under a adversarial attack. Section 4 presents our procedure to certify $(\alpha, \zeta)$ machine learning model safety; it describes the algorithm to certify $(\alpha, \zeta)$ machine learning model safety; and it also presents its guarantees. Section 5 offers a number of experimental results to benchmark $(\alpha, \zeta)$ safety of various machine learning models under various attacks. Finally, we offer various concluding remarks in Section 6. The proofs of the main technical results are relegated to the Supplementary Material.

## 2 RELATED RESEARCH

Our work connects to various research directions in the literature as follows;

**Adversarial Robustness Certification:** There are three major approaches to certify the adversarial robustness of machine learning models (Li et al., 2023): a) set propagation methods (Wong & Kolter, 2018; Wong et al., 2018; Gowal et al., 2018; 2019; Zhang et al., 2019); b) Lipschitz constant controlling methods (Hein & Andriushchenko, 2017; Tsuzuku et al., 2018; Trockman & Kolter, 2020; Leino et al., 2021; Zhang et al., 2021; Xu et al., 2022); and c) randomized smoothing techniques (Cohen et al., 2019; Lecuyer et al., 2019; Salman et al., 2019; Carlini et al., 2023). Set propagation approaches need access to the model architecture and parameters so that an input polytope can be propagated from the input layer to the output layer to produce an upper bound for the worse-case input perturbation. This approach however requires the model architecture to be able to propagate sets, e.g. (Wong & Kolter, 2018) relies on RelU activation functions.Lipschitz constant controlling approaches produce adversarial robustness certification by bounding local Lipschitz constants; however, these approaches are also limited to certain model architectures such as LipConvnet (Singla & Feizi, 2021). In contrast, randomized smoothing (RS) represents a versatile certification methodology free from model architectural constraints or model parameters access.Nonetheless, RS is limited to certifying empirical risk of a machine learning model on pre-defined test datasets under $l_2$-norm bounded adversarial perturbations.Our certification framework shares RS's versatility but a) it also exhibits the ability to accommodate a diverse range of $l_p$ norm-based adversarial perturbations; b) it is not restricted to particular model architectures; and c) it produces a certification for *population* adversarial risk of the machine learning model.

**Other Certification Approaches:** There are also various other recent approaches to certify (audit) machine learning models in relation to issues such as fairness / bias (Black et al., 2020; Xue et al., 2020; Si et al., 2021; Taskesen et al., 2021; Chugg et al., 2023). For example, Black et al. (2020), Xue et al. (2020), Taskesen et al. (2021) and Si et al. (2021) leverage hypothesis testing techniques – coupled with optimal transport approaches – to test whether or not a model discriminates against different demographic groups; Chugg et al. (2023) in turn leverages recent advances in (sequential) hypothesis testing techniques – the "testing by betting" framework – to continuously test (monitor) whether or not a model is fair. Our certification framework also leverages hypothesis testing techniques, but the focus is on certifying for model adversarial robustness in *lieu* of model fairness.

**Distribution-free uncertainty quantification:** Our certification framework builds upon recent work on distribution-free risk quantification, e.g. (Bates et al., 2021; Angelopoulos et al., 2021). In particular, (Bates et al., 2021; Angelopoulos et al., 2021) seek to identify model hyper-parameter configurations that offer a pre-specified level of risk control (under a variety of risk functions). See also similar follow-up work in (Laufer-Goldshtein et al., 2023), (Quach et al., 2023). Our proposed PROSAC framework departs from these existing frameworks since it seeks to offer risk guarantees for a machine learning model in the presence of an adversarial attack: therefore, via the use of a GP-UCB algorithm, it seeks to ascertain one can control the risk of the machine learning model in the presence of the worst-case attacker hyper-parameter configuration.

| Attack Type | Attack Method | Hyperparameter | Hyperparameter Selection |
|---|---|---|---|
| Black-box attack | NES (Ilyas et al., 2018) | $\lambda_\sigma$: forward step size of Gaussian sampling $\lambda_\eta$: step size of input image updating | Empirical |
| | SquareAttack (Andriushchenko et al., 2020) | N.A. | Default |
| White-box attack | AutoAttack (Croce & Hein, 2020b) | N.A. | Default |

Table 1: Representative Black-Box and White-Box attacks, their hyper-parameters, and the hyper-parameter selection procedure. We test two black-box attacks (NES and SquareAttack) and one white-box attack (Auto Attack) to showcase PROSAC's efficacy.

## 3 PROBLEM STATEMENT

We consider how to certify the robustness of a (classification) machine learning model against specific adversarial attacks. We assume that we have access to a machine learning model $\mathcal{M} : \mathcal{X} \to \mathcal{Y}$ that maps features $X \in \mathcal{X}$ onto a (categorical) target $Y \in \mathcal{Y}$ where $(X, Y)$ are drawn from a (unknown) distribution $\mathcal{D}_{X,Y}$. We also assume that this machine learning model has already been optimized (trained) *a priori* to solve a specific multi-class classification task using a given training set (hence, $\mathcal{Y} = \{1, 2, \ldots, K\}$).

We consider that the machine learning model $\mathcal{M}$ is attacked by an adversarial attack $\mathcal{A}_\mathcal{M} : \mathcal{X} \times \mathcal{Y} \to \mathcal{X}$ that given a pair $(X, Y) \in \mathcal{X} \times \mathcal{Y}$ (ideally) converts the original model input $X \in \mathcal{X}$ onto an adversarial one $\tilde{X} \in \mathcal{X}$ as follows:

$$\tilde{X} = \mathcal{A}_\mathcal{M}(X, Y) = X + \tilde{\delta} = X + \arg\max_{\delta \in \mathcal{B}_\epsilon^q} \mathcal{L}(\mathcal{M}(X + \delta), Y), \tag{1}$$

with the intent of maximizing the per-sample loss $\mathcal{L}$ associated with a given sample $(X, Y) \in \mathcal{X} \times \mathcal{Y}$, where $\mathcal{B}_\epsilon^q$ is an $l_q$ norm bounded ball with radius $\epsilon$ (where $\epsilon$ measures the capability of the attacker).

In general, we can distinguish between *white-box* adversarial attacks, where the attacker has access to the machine learning model architecture / parameters, and *black-box* ones, where the attacker does not have access to the machine learning model details. [1]

**White-box attacks.** The most widely used white-box attack is the projected gradient descent (PGD) attack (Madry et al., 2018), where the attacker relies on the signed gradient of the loss with respect to the input to update $\tilde{X}$ iteratively as follows:

$$\tilde{X}^{t+1} = \prod_{X + \mathcal{B}_\epsilon} [\tilde{X}^t + \lambda_\eta \cdot \text{sign}(\nabla_X \mathcal{L}(\mathcal{M}(\tilde{X}^t), Y))], \tag{2}$$

where $t$ represents the $t$-th step of PGD iteration, $\lambda_\eta$ is the step size of each update step, and $X^0$ can be set to be equal to the original image $X$ or the original image plus some random noise. Note we also project the result of each gradient update step onto a $\ell_q$-ball with radius $\epsilon$ centered at $X$.

The hyperparameter $\lambda_\eta$ is often selected heuristically based on a pre-specified dataset like ImageNet (Deng et al., 2009), via random or grid search for instance. To circumvent the difficulty of hyperparameter search, AutoAttack (Croce & Hein, 2020b) has been proposed to automatically select hyperparameters of PGD and fix hyperparameters of three other adversarial attacks, i.e., targeted APGD-DLR (Croce & Hein, 2020b), targeted fast adaptive boundary attack (Croce & Hein, 2020a) and Square Attack (Andriushchenko et al., 2020)) according to common practice, which has become the standard benchmark in the field of white-box adversarial robustness.

**Black-box attacks.** There are generally two categories of black-box attacks: score-based (Andriushchenko et al., 2020) and decision-based (Chen et al., 2020). The idea of score-based attack is to approximate the gradient of the loss with respect to input using zero-order information since the exact differentiation cannot be done without the knowledge of the model parameters. Natural evolution strategy (NES) (Ilyas et al., 2018) is a widely used score-based attack involving two iterative

---

[1]Note that one needs knowledge of the model architecture / parameters to directly calculate the gradient of the loss with respect to the input, in order to optimize the perturbation appearing in equation 1. White-box attacks can indeed compute such a gradient directly, but black-box attacks rely on other approaches.

steps that rely on two crucial hyperparameters, $\lambda_\sigma$ and $\lambda_\eta$. In the first step of iteration $t$, we estimate the gradient of the loss with respect to the input using the natural evolution strategy by relying on $S$ samples from a multi-variate Gaussian distribution, i.e.,

$$G^t = \frac{1}{2S\lambda_\sigma} \sum_{s=1}^{S} [\mathcal{L}_{C\&W}(X^t + \lambda_\sigma u_s, Y) - \mathcal{L}_{C\&W}(X^t - \lambda_\sigma u_s, Y)] \cdot u_s, \tag{3}$$

$$u_s \sim \mathcal{N}(\mathbf{0}, \boldsymbol{I}), \tag{4}$$

where $\mathcal{L}_{C\&W}$ denotes C&W loss (Carlini & Wagner, 2017). In the second step of iteration $t$, we update $\tilde{X}$ using the estimated gradient by relying on projected gradient descent with step size $\lambda_\eta$, i.e.

$$\tilde{X}^{t+1} = \prod_{X+\mathcal{B}_\epsilon} [\tilde{X}^t + \lambda_\eta G^t]. \tag{5}$$

The hyperparameters of NES attack are determined in a heuristic way (Ilyas et al., 2018). In this work, we test our framework with two score-based attacks, NES and SquareAttack (Andriushchenko et al., 2020). The SquareAttack has a hyperparameter that determines the initial value for the attacked image, which is also empirically selected. We will be assuming in the sequel, where appropriate, that that attacker draws its hyper-parameters configuration $\lambda$ from a (finite) set of hyper-parameter configurations $\Lambda$, where each hyper-parameter configuration is $d$-dimensional i.e. $\lambda \in \mathbb{R}^d$ We summarize the adversarial attacks used in our experiments in Tab. 1.

In general, the various attacks are stochastic, i.e., in contrast with equation 1, the white-box and black-box attacks in Table 1 do not deliver a deterministic perturbation $\tilde{\delta}$ given fixed $(X, Y)$ (and given fixed attack hyper-parameters) but rather a random one because the attacks also depend on other random variables. Notably, the white-box PGD attack depends on the initialization $X^0$; the black-box NES attack also depends on the exact samples of the multi-variate Gaussian random variables per iteration; likewise, other black-box attacks also depend on various random quantities like box sampling in SquareAttack. Therefore, we will be representing in the sequel the adversarial attacks as $\mathcal{A}_{\mathcal{M}, \mathcal{B}_\epsilon^q, \lambda}(X, Y, Z)$ to emphasize that its operation also depends on a random object $Z$ drawn from a distribution $\mathcal{D}_Z$, a series of attack hyper-parameters $\lambda = (\lambda_1, \cdots, \lambda_d) \in \Lambda$, the attack budget $\epsilon$ and norm $q$, and naturally the machine learning model $\mathcal{M}$.

Therefore, given an adversarial attack $\mathcal{A}_{\mathcal{M}, \mathcal{B}_\epsilon^q, \lambda}$, we can characterize the performance of the machine learning model using two quantities: the *adversarial risk* and the *max adversarial risk*. We define the adversarial (population) risk induced by the attack $\mathcal{A}_{\mathcal{M}, \mathcal{B}_\epsilon^q, \lambda}$ on the model $\mathcal{M}$ as follows:

$$\mathcal{R}_{\mathcal{A}_{\mathcal{M}, \mathcal{B}_\epsilon^q, \lambda}}(\mathcal{M}) = \mathbb{E}_{(X,Y,Z) \sim \mathcal{D}_{X,Y} \times \mathcal{D}_Z} \left\{ \mathbb{1}[\mathcal{M}(\mathcal{A}_{\mathcal{M}, \mathcal{B}_\epsilon^q, \lambda}(X, Y, Z)) \neq Y] \cdot \mathbb{1}[\mathcal{M}(X) = Y] \right\} \tag{6}$$

and we define the max adversarial (population) risk induced by the attack $\mathcal{A}_{\mathcal{M}, \mathcal{B}_\epsilon^q, \lambda}$ on the model $\mathcal{M}$ independently of how the attacker chooses its hyper-parameters as follows:

$$\mathcal{R}^*_{\mathcal{A}_{\mathcal{M}, \mathcal{B}_\epsilon^q, \lambda}}(\mathcal{M}) = \max_{\lambda \in \Lambda} \mathbb{E}_{(X,Y,Z) \sim \mathcal{D}_{X,Y} \times \mathcal{D}_Z} \left\{ \mathbb{1}[\mathcal{M}(\mathcal{A}_{\mathcal{M}, \mathcal{B}_\epsilon^q, \lambda}(X, Y, Z)) \neq Y] \cdot \mathbb{1}[\mathcal{M}(X) = Y] \right\} \tag{7}$$

where we use the 0-1 loss to measure the per-sample loss. [2] Note that the adversarial (population) risk characterizes the performance of the machine learning model for a specific attack with a given budget / norm, for a fixed hyper-parameters configuration, whereas the max adversarial (population) risk characterizes the performance of the machine learning model for an attack with a given budget / norm, independently on how the attacker chooses its hyper-parameters configuration.

Our overarching goal is to ascertain whether the machine learning model is safe by establishing whether the max (adversarial) population risk is below some threshold with high probability.

**Definition 1.** $((\alpha, \zeta)$-Model Safety) *Fix* $0 \leq \alpha \leq 1$, $0 \leq \zeta \leq 1$. *Then, we say that a machine learning model* $\mathcal{M}$ *is* $(\alpha, \zeta)$-*safe under an adversarial attack* $\mathcal{A}_{\mathcal{M}, \mathcal{B}_\epsilon^q, \lambda}$ *with fixed budget* $\epsilon$ *and norm* $q$, *and for all attack hyper-parameters, provided that*

$$\mathbb{P}\left( \text{reject } \mathcal{R}^*_{\mathcal{A}_{\mathcal{M}, \mathcal{B}_\epsilon^q, \lambda}}(\mathcal{M}) > \alpha \mid \mathcal{R}^*_{\mathcal{A}_{\mathcal{M}, \mathcal{B}_\epsilon^q, \lambda}}(\mathcal{M}) > \alpha \text{ is true} \right) \leq \zeta \tag{8}$$

---

[2] This work concentrates primarily on classification problems with the 0-1 loss. However, our work readily extends to other losses subject to some immediate modifications.

We will see in the sequel this entails formulating an hypothesis testing problem where the null hypothesis is associated with a max adversarial risk higher than $\alpha$. Therefore, $(\alpha, \zeta)$-model safety entails that we declare the model max adversarial risk is less than $\alpha$ when it is in fact higher than $\alpha$ with probability smaller than $\zeta$, or, more loosely speaking, the model max adversarial risk is less than $\alpha$ with probability higher than $1 - \zeta$

## 4 CERTIFICATION PROCEDURE

We now describe our proposed certification approach allowing us to establish $(\alpha, \zeta)$- safety of a machine learning model in the presence of an adversarial attack. We will omit the dependency of the adversarial risks on the model, the attack, and the attack parameters in order to simplify notation. We will also omit that the attack depends on the model, its budget / norm, and the hyper-parameters.

### 4.1 PROCEDURE

Our procedure connects but also departs from a recent line of research relating to risk control in machine learning models, pursued by (Bates et al., 2021; Angelopoulos et al., 2021; Laufer-Goldshtein et al., 2023) (see also references therein). In particular, (Bates et al., 2021; Angelopoulos et al., 2021; Laufer-Goldshtein et al., 2023) offer a methodology to identify a set of model hyper-parameter configurations that control the (statistical) risk of the machine learning model. However, we are not interested in determining a set of attacker hyper-parameters guaranteeing risk control, but rather in guaranteeing risk control independently on how the attacker chooses the hyper-parameters (since the user cannot control how the attacker chooses the hyper-parameters).

Fix the machine learning model $\mathcal{M}$ Fix the adversarial attack $\mathcal{A}$, the adversarial attack budget $\epsilon$, and the adversarial attack norm $q$. [3] We leverage – in line with (Bates et al., 2021; Angelopoulos et al., 2021; Laufer-Goldshtein et al., 2023) – access to a calibration set $\mathcal{S} = \{(X_1, Y_1), (X_2, Y_2), \ldots, (X_n, Y_n)\}$ (independent of any training set) where the samples $(X_i, Y_i)$ are drawn i.i.d. from the distribution $\mathcal{D}_{X,Y}$ to construct our certification procedure.

Our certification procedure then involves the following sequence of steps:

- First, we set up an hypothesis testing problem where the null hypothesis is $\mathcal{H}_0 : \mathcal{R}^* > \alpha$ or, equivalently, $\mathcal{H}_0 : \exists \lambda : \mathcal{R}(\lambda) > \alpha$. [4]

- Second, we leverage the calibration set (plus another set with a number of instances / objects characterizing the randomness of the attack) to determine a finite-sample $p$-value $p^*$ that can be used for accepting or rejecting the null hypothesis $\mathcal{H}_0 : \mathcal{R}^* > \alpha$ or, equivalently, $\mathcal{H}_0 : \exists \lambda : \mathcal{R}(\lambda) > \alpha$.

- Finally, we reject or accept the null hypothesis depending on whether or not the $p$-value $p^*$ is less than or greater than $\zeta$, respectively.

This procedure allows us to immediately establish $(\alpha, \zeta)$- safety of the machine learning model $\mathcal{M}$ in the presence of an adversarial attack $\mathcal{A}$, in accordance with Definition 1.

**Theorem 1.** *Let $p^*$ be a p-value associated with the hypothesis testing problem where the null hypothesis is $\mathcal{H}_0 : \mathcal{R}^* > \alpha$ or, equivalently, $\mathcal{H}_0 : \exists \lambda : \mathcal{R}(\lambda) > \alpha$. It follows immediately that the machine learning model is $(\alpha, \zeta)$- safe, i.e.*

$$\mathbb{P}\left(\text{reject } \mathcal{R}^* > \alpha \mid \mathcal{R}^* > \alpha \text{ is true}\right) \leq \zeta \tag{9}$$

We next show how to derive a $p$-value for our hypothesis testing problem where $\mathcal{H}_0 : \exists \lambda : \mathcal{R}(\lambda) > \alpha$ from the $p$-values for the hypotheses testing problems where $\mathcal{H}_0 : \mathcal{R}(\lambda) > \alpha, \forall \lambda$ (see also (Laufer-Goldshtein et al., 2023)). [5]

---

[3] We do not consider the attack budget and norm to be hyper-parameters; indeed, it would not be possible to control the risk where the adversary has the ability to choose any attack budget $\epsilon \in (0, \infty)$

[4] $\mathcal{R}^*$ represents the max adversarial risk in equation 7 and $\mathcal{R}(\lambda)$ represents the adversarial risk in equation 6 where we emphasize it depends on the attacker hyper-parameters $\lambda \in \Lambda$.

[5] We emphasize the difference between the hypotheses testing problems. The hypothesis testing problem with null $\mathcal{H}_0 : \exists \lambda : \mathcal{R}(\lambda) > \alpha$ tests whether the max adversarial risk is above $\alpha$ independently of the choice of

---

**Algorithm 1** GP-UCB for hyperparameter optimization

---

**Input:** Prior $GP(0, k)$, parameters $\beta$.
**for** t = 1, 2, 3 ... T **do**
 Choose $\lambda_t = \arg\max_{\lambda \in \Lambda} \mu_{t-1}(\lambda) + \beta \sigma_{t-1}(\lambda)$.
 Observe reward $\hat{p}_t = p(\lambda_t) + \epsilon_t$.
 Perform update to get a new GP using the sampled point $(\lambda_t, \hat{p}_t)$.
**end for**
**return** $\hat{p}_T = 1/T \sum_{t=1}^{T} \hat{p}_t$

---

**Theorem 2.** *If $p(\lambda)$ is a p-value associated with the null $\mathcal{H}_0 : \mathcal{R}(\lambda) > \alpha$ then $p^* = \max_{\lambda \in \Lambda} p(\lambda)$ is a p-value associated with the null hypothesis $\mathcal{H}_0 : \exists \, \lambda, \mathcal{R}(\lambda) > \alpha$.*

Therefore, building upon Theorem 2, we can immediately determine a $p$-value for our hypothesis testing problem.

**Theorem 3.** *A (super-uniform) p-value associated with the null hypothesis $\mathcal{H}_0 : \exists \, \lambda, \mathcal{R}(\lambda) > \alpha$ is given by:*

$$p^* = \max_{\lambda \in \Lambda} \min \left\{ \exp\left( -n \cdot h_1\left( \hat{\mathcal{R}}(\lambda) \wedge \alpha, \alpha \right) \right), e \cdot \mathbb{P}\left( \mathsf{Bin}(n, \alpha) \leq \left\lceil n \cdot \hat{\mathcal{R}}(\lambda) \right\rceil \right) \right\} \quad (10)$$

*where $\hat{\mathcal{R}}(\lambda)$ represents the adversarial empirical risk induced by the attack $\mathcal{A}$ on model $\mathcal{M}$ given a specific hyper-parameter configuration $\lambda \in \Lambda$ i.e.*

$$\hat{\mathcal{R}}(\lambda) = \frac{1}{n} \sum_{i=1}^{n} \mathbb{1}[\mathcal{M}(\mathcal{A}(X_i, Y_i, Z_i)) \neq Y_i] \cdot \mathbb{1}[\mathcal{M}(X_i) = Y_i] \quad (11)$$

*where (again) $\mathcal{S} = \{(X_1, Y_1), \ldots, (X_n, Y_n)\}$ is the set containing the calibration data, $\mathcal{Z} = \{Z_1, \ldots, Z_n\}$ is a set containing a series of random objects that capture the randomness of the attack, and $h_1(a, b) = a \cdot \log(a/b) + (1 - a) \cdot \log((1 - a)/(1 - b))$.*

## 4.2 ALGORITHM AND ITS GUARANTEES

Our procedure to establish $(\alpha, \zeta)$-safety of a learning-based model $\mathcal{M}$ in the presence of an adversarial attack $\mathcal{A}$, in accordance with Definition 1, relies on the ability to approximate the $p$-value associated with the null hypothesis $\mathcal{H}_0 : \exists \, \lambda : \mathcal{R}(\lambda) > \alpha$ as per Theorem 3. However, this involves solving a complex optimization problem over the set of attacker hyper-parameter configurations.

We therefore propose to adopt a Bayesian optimization (BO) procedure, based on the established Gaussian Process Upper Confidence Bound (GP-UCB) algorithm (Srinivas et al., 2010), since it can be used to effectively search over the set of hyper-parameter configurations of the attack and consequently identify the configuration leading to the highest $p$-value. We require a sample-efficient optimization method since the evaluation of the $p$-value involves computation of the empirical risk of the model subject to the attack which is known to be time-consuming for large and complex models such as ViT-Large used in our experiments. Algorithm 1 summarizes our algorithm to search for the attack hyperparameters.

The following theorem shows that we can still establish $(\alpha, \zeta)$-safety of the machine learning model $\mathcal{M}$ in the presence of an adversarial attack $\mathcal{A}$ (in accordance with Definition 1), provided that the number of rounds (i.e., samples) of the GP-UCB algorithm in Algorithm 1 is sufficiently large.

**Theorem 4.** *($(\alpha, \zeta)$-Model Safety with GP-UCB) Fix $0 \leq \alpha \leq 1$, $0 \leq \zeta \leq 1$, the machine learning model $\mathcal{M}$, the adversarial attack $\mathcal{A}$ (its budget $\epsilon$ and norm q). Then, we can guarantee that the machine learning model $\mathcal{M}$ is $(\alpha, \zeta)$-safe under an adversarial attack $\mathcal{A}$ for all attack hyper-parameters, i.e.,*

$$\mathbb{P}\left( \text{reject } \mathcal{R}^* > \alpha \mid \mathcal{R}^* > \alpha \text{ is true} \right) \leq \zeta, \quad (12)$$

*by relying on Algorithm 1 – with a suitable number of rounds – to approximate the p-values required by our procedure.*

---

hyper-parameters associated with the attack, whereas the hypothesis testing problem with null $\mathcal{H}_0 : \mathcal{R}(\lambda) > \alpha$ tests whether the risk is above $\alpha$ for a particular choice of hyper-parameters associated with the attack.

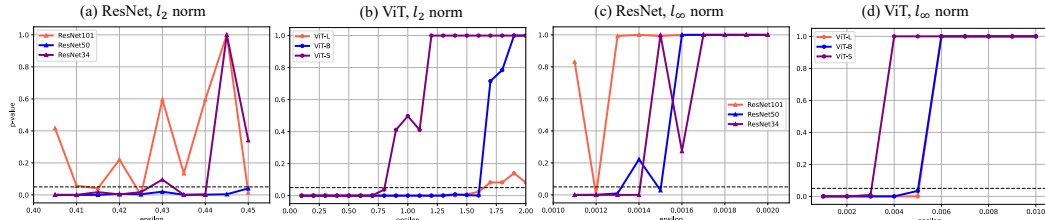

Figure 1: Adversarial risk certification for various models under AutoAttack with $l_2$ norm and $l_\infty$ norm.

Figure 2: Adversarial risk certification for various models under SquareAttack with $l_2$ norm and $l_\infty$ norm.

The GP UCB algorithm in Algorithm 1 delivers a $p$-value estimate that is close to the true $p$-value with probability $1 - \delta$ only, therefore the testing procedure underlying Theorem 4 compares the GP UCB $p$-value estimate to a more conservative threshold $0 < \zeta' < \zeta$, rather than $\zeta$, in order to retain a type-I error probability bound akin to that in Definition 1. This is possible by making sure the number of GP UCB rounds is sufficiently large. See Supplementary Material.

## 5 EXPERIMENTS

We now showcase how to use PROSAC to certify the performance of various state-of-the-art vision models in the presence of a variety of adversarial attacks.

### 5.1 EXPERIMENTAL SETTINGS

**Datasets** We follow the common experimental setting in black-box adversarial attacks, using 1,000 images from ImageNet (Andriushchenko et al., 2020; Ilyas et al., 2018) to apply our proposed certification procedure. In particular, we take our calibration set to correspond to this dataset.

**Models** We use two representative state-of-the-art models in computer vision, i.e., vision transformer (ViT) (Dosovitskiy et al., 2020) and ResNet (He et al., 2016), in our experiments. To make a comparison between models of different sizes, we use small, base and large models for both model architectures. Specifically, we tested ViT-Small, ViT-Base and ViT-Large for ViT, and ResNet-34, ResNet-50 and ResNet-101 for ResNet.

**Adversarial Attacks** We also use three adversarial attacks in our experiment, including one white-box attack and two black-box ones. The AutoAttack (Croce & Hein, 2020b) is used to evaluate the white-box adversarial risk as it is the default benchmark for white-box adversarial robustness in literature. We use SquareAttack (Andriushchenko et al., 2020) and NES attack (Ilyas et al., 2018) in the black-box setting, as both attacks are computational efficient and effective. Both attacks are score-based while the SquareAttack is considered as hyperparameter-free and the NES attack contains two hyperparameters $\lambda_\sigma$ and $\lambda_\eta$. We also use both $l_2$- and $l_\infty$-balls with radius $\epsilon$ to define the various attacks. We use $\alpha = 0.10$ and $\zeta = 0.05$ in the safety certification.

### 5.2 EXPERIMENTAL RESULTS

We now report on various results relating to the use of PROSAC to certify the performance of the various models in the presence of the various attacks, including those with fixed and those with optimizable hyper-parameters.

**Adversarial Attacks with Fixed Hyperparameters.** Here, we consider SquareAttack and AutoAttack by fixing their hyperparameters to be equal to the default ones, so it suffices to certify the machine learning models for different values of attack budgets and different norms. Fig. 1 depicts

| (a) Sparse Grid | | | | | |
|---|---|---|---|---|---|
| $\epsilon$ | 0.10 | 0.15 | 0.2 | 0.25 | 0.3 |
| p-value | 0.000 | 0.009 | 0.423 | 1.000 | 1.000 |

| (b) Dense Grid | | | | | |
|---|---|---|---|---|---|
| $\epsilon$ | 0.14 | 0.15 | 0.16 | 0.17 | 0.18 |
| p-value | 0.000 | 0.009 | 0.029 | 0.134 | 0.346 |

Table 2: NES attack with GP-UCB optimization (Alg. 1) for hyperparameter selection with ResNet50 and $l_2$ norm.

how the $p$-value behaves versus attack budget for an AutoAttack, with hyperparameter set to be equal to the default one inCroce & Hein (2020b), for the different machine learning models. In particular, we let the attack budget lie in the grid $\epsilon = \{0.021, 0.022, \cdots, 0.030\}$ for $l_2$-norm constrained perturbations and in the grid $\epsilon = \{\text{1e-5}, \cdots, \text{1e-4}\}$ for $l_\infty$-norm constrained perturbations, where we choose smaller adversarial attack budget for $l_\infty$ constrained attacks as an $l_2$ ball of a certain radius is contained by the $l_\infty$ ball with same radius. Fig. 2 depicts how the $p$-value behaves versus attack budget for an SquareAttack, where we have set the hyperparameter corresponding to the probability of changing a particular image pixel to be equal to the default one of 0.05 c.f. Andriushchenko et al. (2020). Note that ResNets and ViTs exhibit radically different behavious under a SquareAttack, so we also use different attack budget grids for these two different models

**Adversarial Attacks with Free Hyperparameters.** Here, we consider instead a NES attack where the attacker can choose the two hyperparameters $\lambda_\sigma$ and $\lambda_\eta$ shown in Tab. 1, in order to test the ability of the BO algorithm to certify machine learning model robustness. The Bayesian optimization is initialized with 9 initial samples using a two-dimension discrete grid, where $\lambda_\sigma = \{0.005, 0.01, 0.015\}$ and $\lambda_\eta = \{0.01, 0.02, 0.03\}$. During the GP-UCB optimization process, we set $\beta_{\text{UCB}}$=0.1, the interval bound for both hyperparameters $[\text{1e-5}, 0.1]$ and the number of optimization rounds $T = 50$. For the attack budget, we use a grid of $\{0.10, 0.15, 0.20, 0.25, 0.30\}$. Tab 2 showcases the $p$-value for different $\ell_2$-norm constrained attack budgets for the ResNet50 model.

**Discussion.** Our experimental results reveal various findings. First, we observe that ViTs are generally more adversarially robust than ResNets under both white-box and black-box attacks, corroborating existing observations in (Shao et al., 2022; Bhojanapalli et al., 2021). For instance, in both Fig. 1 and 2, ViT-B and ViT-L are certifiably more robust than all ResNets under both attacks. Second, we also observe that larger ViT models appear to be more robust than smaller ones. In contrast, the Resnet model size does not appear to influence much its robust against adversarial attacks, in line with existing research work suggesting that a wider ResNet does not necessarily have a stronger adversarial robustness (Wu et al., 2021). Third, we also note that a given model exhibits completely different certifiable robustness in the presence of different adversarial attacks. It is clear from Fig. 1 and Fig. 2 that – for a specific attack budget and norm – it is more difficult to guarantee model safety in the presence of the white-box AutoAttack in comparison with the black-box SquareAttack. Moreover, in the presence of NES attack where the attacker can also optimize their attack hyperparameters, it is also more difficult to ensure ($\alpha = 0.10, \zeta = 0.05$) model safety in comparison with the SquareAttack (e.g. we can certify the ResNet50 is ($\alpha, \zeta$)-safe with $\epsilon = 0.4$ in the presence of an $\ell_2$-norm based SquareAttack but not in the presence of an $\ell_2$-norm based NES attack. Finally, we remark that – due to the stochasticity of the attacks – the $p$-values do not always monotonically increase with the attack budget; interestingly, this issue is particularly accute with the SquareAttack (since it involves attacking a fraction of the image pixels), implying that it is virtually impossible to certify ($\alpha = 0.10, \zeta = 0.05$) model safety for certain models such as ResNet50 and 101.

# 6 CONCLUSIONS

We have proposed PROSAC, a new approach to certify the performance of a machine learning model in the presence of an adversarial attack, with population level adversarial risk guarantees. PROSAC builds on recent work on distribution-free risk quantification approaches, offering an instrument to ascertain whether a model is likely to be safe in the presence of an adversarial attack, independently of how the attacker chooses the attack hyperparameters. We show via experiments that PROSAC is able to certify various state-of-the-art models, leading to results that are in line with existing results in the literature. PROSAC has also unveiled that large ViT models appear to be more adversarially robust than smaller ones, pointing to new directions for research relating to the relationship between the capacity of a ViT and its adversarial robustness. The technical framework developed here is likely to be of high relevance to AI regulation, such as the EU's proposed AI Act, which requires providers of certain AI systems to ensure that their systems are resilient to adversarial attacks. Our approach to certifying the performance of any black-box machine learning system offers a tool that can help providers to discharge their legal obligations and show that they acted with due diligence.

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

## A  PROOF OF THEOREM 1

The proof is trivial: We reject / accept the null hypothesis $\mathcal{H}_0 : \mathcal{R}^* > \alpha$ depending on whether or not $p^* \leq \zeta$, respectively. Then, the result follows immediately because $p^*$ is a finite-sample valid $p$-value under the null, i.e. $\mathbb{P}\left(p^* \leq \zeta\right) \leq \zeta$ under the null ($0 \leq \zeta \leq 1$).

## B  PROOF OF THEOREM 2

The proof is also in Laufer-Goldshtein et al. (2023). In particular, we can establish that

$$\mathbb{P}\left(p^* \leq \zeta\right) = \mathbb{P}\left(\max_{\lambda \in \Lambda} p(\lambda) \leq \zeta\right) = \mathbb{P}\left(p \leq \zeta, \forall\, \lambda \in \Lambda\right) \leq \max_{\lambda \in \Lambda} \mathbb{P}\left(p(\lambda) \leq \zeta\right) \leq \zeta \tag{13}$$

where the last step follows from the fact that $\mathbb{P}\left(p(\lambda) \leq \zeta\right) \leq \zeta, \forall\, \lambda \in \Lambda$. Therefore, $p^* = \max_{\lambda \in \Lambda} p(\lambda)$ is a (super-uniform) p-value associated with the null hypothesis $\mathcal{H}_0 : \exists\, \lambda, \mathcal{R}(\lambda) > \alpha$.

## C  PROOF OF THEOREM 3

The proof follows immediately from (Bates et al., 2021) with some very minor modifications that accommodate for the fact that the attack can be stochastic.

Fix the attacker hyper-parameter configuration $\lambda \in \Lambda$. We can show based on the tighter version of Hoeffding's inequality (Hoeffding, 1994) that for any $\mathcal{R}(\lambda) > \alpha$ it holds

$$\mathbb{P}\left(\hat{\mathcal{R}}(\lambda) \leq \alpha\right) \leq \exp\left(-n \cdot h_1\left(\alpha; \mathcal{R}(\lambda)\right)\right) \tag{14}$$

We can also show based on Bentkus inequality that it holds

$$\mathbb{P}\left(\hat{\mathcal{R}}(\lambda) \leq \alpha\right) \leq e \cdot \mathbb{P}\left(\mathrm{Bin}(n, \mathcal{R}(\lambda)) \leq \lceil n \cdot \alpha \rceil\right) \tag{15}$$

Therefore, via the hybridization of the Hoeffding and Bentkus inequalities (Bates et al., 2021) it also follows that

$$\mathbb{P}\left(\hat{\mathcal{R}}(\lambda) \leq \alpha\right) \leq \min\left\{\exp\left(-n \cdot h_1\left(\alpha; \mathcal{R}(\lambda)\right)\right), e \cdot \mathbb{P}\left(\mathrm{Bin}(n, \mathcal{R}(\lambda)) \leq \lceil n \cdot \alpha \rceil\right)\right\} \tag{16}$$

implying that (Bates et al., 2021)

$$p(\lambda) = \min\left\{\exp\left(-n \cdot h_1\left(\hat{\mathcal{R}}(\lambda) \wedge \alpha, \alpha\right)\right), e \cdot \mathbb{P}\left(\mathrm{Bin}(n, \alpha) \leq \left\lceil n \cdot \hat{\mathcal{R}}(\lambda) \right\rceil\right)\right\} \tag{17}$$

is a valid $p$-value associated with the null hypothesis $\mathcal{H}_0 : \mathcal{R}(\lambda) > \alpha$ and – via Theorem 3

$$p^* = \max_{\lambda \in \Lambda} \min\left\{\exp\left(-n \cdot h_1\left(\hat{\mathcal{R}}(\lambda) \wedge \alpha, \alpha\right)\right), e \cdot \mathbb{P}\left(\mathrm{Bin}(n, \alpha) \leq \left\lceil n \cdot \hat{\mathcal{R}}(\lambda) \right\rceil\right)\right\} \tag{18}$$

is a valid $p$-value associated with the null hypothesis $\mathcal{H}_0 : \exists \lambda : \mathcal{R}(\lambda) > \alpha$.

## D  PROOF OF THEOREM 4

The proof builds upon a classical result establishing regret bounds for Gaussian Process Upper Confidence Bound (GP-UCB) optimization from (Chowdhury & Gopalan, 2017). It is assumed that the p-value lies in an RKHS $\mathcal{H}_k$ with some known kernel $k$, such that $\|p\|_k \leq B$, and that the noise sequence is conditionally $R$-sub-Gaussian, as in (Chowdhury & Gopalan, 2017).

Let $\hat{p}_t$ correspond to the $p$-value evaluation corresponding to the GP-UCB's decision $\lambda_t$ at round $t$, i.e., $\hat{p}_t = p(\lambda_t)$. We let $\hat{p}_T = 1/T \sum_{t=1}^{T} \hat{p}_t$ correspond to the GP-UCB algorithm (maximal) $p$-value approximation (after $T$ rounds) and $p^*$ correspond to the exact (maximal) $p$-value appearing in Theorem 3. We can establish from (Chowdhury & Gopalan, 2017) that with probability at least $1 - \delta$ (with $\delta \in (0, 1)$) [6]

$$p^* - \hat{p}_T \leq \mathcal{O}\left(B\sqrt{\gamma_T/T} + \sqrt{\gamma_T\left(\gamma_T + \log(1/\delta)\right)/T}\right), \tag{19}$$

---

[6]Note that this probability is with respect to the randomness of the noisy observations.

where $\gamma_T$ corresponds to the maximum information gain at round $T$ (Chowdhury & Gopalan, 2017). We can also establish that

$$\hat{p}_T \geq p^* - \mathcal{O}\left(B\sqrt{\gamma_T/T} + \sqrt{\gamma_T\left(\gamma_T + \log(1/\delta)\right)/T}\right) \tag{20}$$

with probability greater than $1 - \delta$, and

$$\hat{p}_T < p^* - \mathcal{O}\left(B\sqrt{\gamma_T/T} + \sqrt{\gamma_T\left(\gamma_T + \log(1/\delta)\right)/T}\right) \tag{21}$$

with probability less than $\delta$.

We now propose to reject or accept the hypothesis $\mathcal{H}_0 : \mathcal{R}^* > \alpha$ by comparing $\hat{p}_T$ to a threshold $\zeta'$ in *lieu* of the original threshold $\zeta$, where we will define the value of the new threshold later, because we can only guarantee that $\hat{p}_T$ is close to $p^*$ – per equation 19 – with probability $1 - \delta$.

We next quantify the probability of rejection of the null hypothesis given the null hypothesis is true. In particular, via the law of total probability, we can show that [7]

$$
\begin{aligned}
\mathbb{P}\left(\hat{p}_T \leq \zeta'\right) = {} & \mathbb{P}\Big(\hat{p}_T \leq \zeta' \mid \hat{p}_T \geq p^* - \mathcal{O}\left(B\sqrt{\gamma_T/T} + \sqrt{\gamma_T\left(\gamma_T + \log(1/\delta)\right)/T}\right)\Big) \times \\
& \times \mathbb{P}\left(\hat{p}_T \geq p^* - \mathcal{O}\left(B\sqrt{\gamma_T/T} + \sqrt{\gamma_T\left(\gamma_T + \log(1/\delta)\right)/T}\right)\right) + \\
= {} & \mathbb{P}\Big(\hat{p}_T \leq \zeta' \mid \hat{p}_T < p^* - \mathcal{O}\left(B\sqrt{\gamma_T/T} + \sqrt{\gamma_T\left(\gamma_T + \log(1/\delta)\right)/T}\right)\Big) \times \\
& \times \mathbb{P}\left(\hat{p}_T < p^* - \mathcal{O}\left(B\sqrt{\gamma_T/T} + \sqrt{\gamma_T\left(\gamma_T + \log(1/\delta)\right)/T}\right)\right)
\end{aligned} \tag{22}
$$

We upper bound the first probability in equation 22 as follows:

$$
\begin{aligned}
\mathbb{P}\Big(\hat{p}_T \leq \zeta' \mid \hat{p}_T \geq {} & p^* - \mathcal{O}\left(B\sqrt{\gamma_T/T} + \sqrt{\gamma_T\left(\gamma_T + \log(1/\delta)\right)/T}\right)\Big) \leq \\
& \leq \mathbb{P}\left(p^* \leq \zeta' + \mathcal{O}\left(B\sqrt{\gamma_T/T} + \sqrt{\gamma_T\left(\gamma_T + \log(1/\delta)\right)/T}\right)\right) \leq \\
& \leq \zeta' + \mathcal{O}\left(B\sqrt{\gamma_T/T} + \sqrt{\gamma_T\left(\gamma_T + \log(1/\delta)\right)/T}\right)
\end{aligned} \tag{23}
$$

because $\hat{p}_T \leq \zeta' \implies p^* \leq \zeta' + \mathcal{O}\left(B\sqrt{\gamma_T/T} + \sqrt{\gamma_T\left(\gamma_T + \log(1/\delta)\right)/T}\right)$ under the condition $\hat{p}_T \leq p^* - \mathcal{O}\left(B\sqrt{\gamma_T/T} + \sqrt{\gamma_T\left(\gamma_T + \log(1/\delta)\right)/T}\right)$.

We trivially upper bound the third probability in equation 22 as follows:

$$\mathbb{P}\Big(\hat{p}_T \leq \zeta' \mid \hat{p}_T < p^* - \mathcal{O}\left(B\sqrt{\gamma_T/T} + \sqrt{\gamma_T\left(\gamma_T + \log(1/\delta)\right)/T}\right)\Big) \leq 1 \tag{24}$$

Furthermore, in view of the probabilistic guarantee associated with the GP UCB algorithm in equation 19, we also upper bound the remaining probabilities as follows:

$$\mathbb{P}\left(\hat{p}_T < p^* - \mathcal{O}\left(B\sqrt{\gamma_T/T} + \sqrt{\gamma_T\left(\gamma_T + \log(1/\delta)\right)/T}\right)\right) \leq 1 \tag{25}$$

and

$$\mathbb{P}\left(\hat{p}_T \geq p^* - \mathcal{O}\left(B\sqrt{\gamma_T/T} + \sqrt{\gamma_T\left(\gamma_T + \log(1/\delta)\right)/T}\right)\right) \leq \delta \tag{26}$$

Putting this together, it follows that – under the null hypothesis – we have that

$$\mathbb{P}\left(\hat{p}_T \leq \zeta'\right) \leq \zeta' + \mathcal{O}\left(B\sqrt{\gamma_T/T} + \sqrt{\gamma_T\left(\gamma_T + \log(1/\delta)\right)/T}\right) + \delta \tag{27}$$

Finally, we guarantee $(\alpha, \zeta)$ model safety by choosing the new threshold $\zeta' = \zeta - \mathcal{O}\left(B\sqrt{\gamma_T/T} + \sqrt{\gamma_T\left(\gamma_T + \log(1/\delta)\right)/T}\right) - \delta$. Note we can guarantee $\zeta > \mathcal{O}\left(B\sqrt{\gamma_T/T} + \sqrt{\gamma_T\left(\gamma_T + \log(1/\delta)\right)/T}\right) + \delta$ by choosing the number of GP UCB rounds to be sufficiently large, for any $\delta < \zeta$.

---

[7]This probability is computed with respect to the randomness of the GP-UCB solution, the randomness of the calibration set, and the randomness of the attack, under the null hypothesis.

