# OpenReview forum: "PROSAC: Provably Safe Certification for Machine Learning Models under Adversarial Attacks"
_ICLR.cc/2024/Conference — ICLR 2024 Conference Withdrawn Submission_

### Official Review · Reviewer_e7ug · 2023-10-28

**Soundness:** 3 good
**Presentation:** 3 good
**Contribution:** 2 fair
**Rating:** 5
**Confidence:** 3

**Summary:**

The authors both propose a new Bayesian mechanism for certifications, and attempt to demonstrate the relative performance of different architectures.

**Strengths:**

The presented framework is interesting, and the introduction as presented takes a very unique perspective on the reasons why there is a critical need for research in the field of AI security.

**Weaknesses:**

While the idea within this work is interesting, I do not believe it has suitable rigorous experimentation (especially in terms of dataset diversity), or experimentation (does not follow standard expectations regarding the trade off between certification proportion and size that are common in other certification papers). While there is validity in a paper that demonstrates that a new approach has the potential to extend the ability of certifications to new frontiers - however, part of doing this kind of validation would require a comprehensive set of experiments demonstrating scaling and performance, all of which are missing.

Also one of the stated contributions of this work is to extend Randomised Smoothing from $\ell_2$ to $\ell_p$. However, this is missing a wide range of literature on $\ell_p$ certifications in randomised smoothing, see Yang et. al "Randomised Smoothing of All Shapes and Sizes", 2020 as an example of this.

I also worry that this paper is attempting to cover quite a few bases - it's trying to both introduce a Bayesian optimisation mechanism and to demonstrate that VIT's are more robust than other architectures. But in attempting to cover both of these points I believe that neither contribution is sufficiently addressed - the Bayesian mechanism is insufficiently detailed, implementation details are sparse, and the range of experiments (including datasets and metrics) are sparse relative to the level of experimental evidence to truly make these points.

As a few other notes:
- Figure 1 talks about "adversarial risk certification for various models under AutoAttack" - the involvement of AutoAttack is only tangentially referred to within the document. Figures 1 & 2 don't even seem to be referenced in text? So there's no
- Just as a note on page 3 there's no space between "functions.Lipschitz" in the sentence relating to Wong & Kolter.
- Citation capitalisation is inconsistent - especially when it comes to the names of journals / conferences / venues.
- The paper could do with algorithms and implementation details, even just in the appendices.

**Questions:**

What's the difference between the $(\alpha, \zeta)$ safety framework relative to something like Differential Privacy (as considered by Lecuyer)?

---

> ### Author Response · Authors · 2023-11-22
> **Response to Reviewer e7ug**
>
> We thank the reviewer for the insightful comments. Here are our responses.
>
> **Q1: I do not believe it has suitable rigorous experimentation (especially in terms of dataset diversity), or experimentation (does not follow standard expectations regarding the trade off between certification proportion and size that are common in other certification papers). While there is validity in a paper that demonstrates that a new approach has the potential to extend the ability of certifications to new frontiers - however, part of doing this kind of validation would require a comprehensive set of experiments demonstrating scaling and performance, all of which are missing.**
>
> R1: We use ImageNet as the dataset as it is one of the default dataset to certify adversarial robustness. There is no explicit trade-off between certification radius and sample sizes in our method, because our method does not need more samples/trials as the radius is increased. If the radius is increased, we run the attack algorithm with that radius and get the empirical risk to compute the p-value. About the scaling and performance, we will consider experiments with a larger sample size in our next version, to illustrate its impact on the certification procedure.
>
> **Q2: Also one of the stated contributions of this work is to extend Randomised Smoothing from $l_2$ to $l_p$. However, this is missing a wide range of literature on $l_p$ certifications in randomised smoothing, see Yang et. al "Randomised Smoothing of All Shapes and Sizes", 2020 as an example of this.**
>
> R2: Thanks for mentioning this. It is worth noting that Yang et. al also has the drawback of only giving a certification for certain test samples, while the major contribution of our method is to certify a model’s adversarial risk at the population level. We will add the discussion with Yang et. al in the next version.
>
> **Q3: I also worry that this paper is attempting to cover quite a few bases - it's trying to both introduce a Bayesian optimisation mechanism and to demonstrate that VIT's are more robust than other architectures.**
>
> R3: Our observation about off-the-shelf ViT’s is based on the result of certifying the ViT’s with our method, so we are not claiming in any way this is a substantial contribution. Our main contribution first and foremost involves the proposal of a certification methodology, including the use of off-the-shelf Bayesian optimization methods and associated theoretical guarantees allowing one to certify a machine learning model in practice. We will revise the paper to address this concern.
>
> **Q4: Figures 1 & 2 don't even seem to be referenced in text?**
>
> R4: They are referenced as Fig. 1 and 2 of page 8.
>
> **Q5: What’s the difference between the proposed safety framework relative to something like Differential Privacy (as considered by Lecuyer)?**
>
> R5: Lecuyer uses differential privacy techniques to certify defenses against adversarial attacks. Their approach leverages differential privacy bounds to implement certified robustness checks for individual predictions. Our approach in turn offers the means to certify the population-level performance of any machine learning model in the presence of an adversarial attack.

---

### Official Review · Reviewer_yeRm · 2023-10-30

**Soundness:** 1 poor
**Presentation:** 1 poor
**Contribution:** 2 fair
**Rating:** 3
**Confidence:** 4

**Summary:**

This work introduces PROSAC, a method to certify a machine learning model's robustness against an adversarial attack type, regardless of the hyperparameters chosen for that adversarial attack. The claims are substantiated by experiments attacking a few vision models with benchmark attacks. This work also applies Guassian Process Upper Confidence Bound (GP-UCB) to hyperparameter selection during the certification process.

**Strengths:**

* (Moderate) The paper is well-motivated in showing the need for robustness certifications as ML models are used for increasingly critical areas and will likely be subject to more government regulations.

**Weaknesses:**

* (Major) The work's presentation overall is difficult for me to understand. This includes the use of undefined variables and terms in the writing and algorithms as well as hard to find experimental details. Details in questions.

* (Major) In certifying the space of attack hyperparameters a model is robust to, it is unclear what hyperparameters are being varied in each attack and what hyperparameter values the work is certifying are safe. Also, this work seems to omit relevant attack hyperparameters from the certification, including attack budget and constraint norm.

* (Major) The experimental results in section 5 seem to suggest the certification is unreliable. For example, Figures 1 and 2 show very different p-values for only slightly different hyperparameter values where I would expect p-values to be similar for similar hyperparameter values. An example of this is Fig 2c showing epsilon 0.0011 having a p-value above 0.8, epsilon of 0.0012 having a p-value of 0.0, then epsilon of 0.0013 having a p-value of 1.

* (Moderate) Section 2 contains some confusing statements about prior work. Details in questions.

* (Moderate) Theorem 4 seems to say that multiple rounds of approximation are needed to certify a machine learning model, but it does not quantify or attempt to make a statement about that number of rounds of approximation. Later in the text, there seems to be a sentence saying "See Supplementary Material" for this information. However, as it is core to this work's claim, at least a summary of the math required to estimate how many rounds should be executed is required.

**Questions:**

* What is the relation between this work and PAC (Probably Approximately Correct) learning / adversarial PAC learning? It seems the guarantees are very similar to those proposed in this work.


* Regarding the results shown in Figures 1 and 2, I would have expected similar values of epsilon to have similar p-values, with a monotonic increase in p-value as epsilon increases. Why is this not the case and why is there so much variability in p-values? These large differences indicates an unreliable certification since a very small decrease or increase of epsilon can change the p-value from 0 to 1 (as between epsilon = 0.0012 and 0.0013 in Fig 2c).


* Section 2 states "... RS (randomized smoothing) is limited  to certifying empirical risk of a machine learning model on pre-defined test datasets under $l_2$-norm bounded adversarial perturbations." What is meant by a pre-defined test dataset and how is randomized smoothing restricted to it? How is PROSAC not restricted to it?

* Section 2 states "randomized smoothing (RS) represents a versatile certification methodology free from model architectural constraints or model parameters access" but then later states "Our certification framework shares RS’s versatility but a) it also exhibits the ability to accommodate a diverse range of lp norm-based adversarial perturbations; b) it is not restricted to particular model architectures...". These statements seem to first state that RS is free from model architectural constraints but then states it is restricted to particular model architectures. Do I misunderstand what is being said?

* In Table 1, why is the hyperparameter field "N.A." for AutoAttack? There are several hyperparameters that can be set (e.g., number of gradient steps, number of expecation over transformation estimates, the type of attack to execute, etc.).


* Equation 10 is difficult for me to understand. What is $h_1$? Could a narrative be given for what this equation is saying?

* In section 4.1, a footnote says that the attack budget and norm are not considered hyper-parameters because it would not be possible to control the risk if the adversary can choose any attack budget. This justification does not explain why the norm is not considered a hyper-parameter. Is there a reason the norm is not considered a hyperparameter?

* In equation 7, shouldn't the risk no longer have a $\lambda$ subscript?

* In the paragraph below equation 5, I don't understand what is meant by "We will be assuming in the sequel, where appropriate,..." and later "We will be representing in the sequel..." What is the meaning of the word sequel here?...

* In algorithm 1, what is $\beta$ and what is $k$? Is $\mu$ and $\sigma$ the mean and variance of $\lambda$?

---

> ### Author Response · Authors · 2023-11-22
> **Response to Reviewer yeRm**
>
> We thank the reviewer for the insightful comments. Here are our responses.
>
> **Q1: The work's presentation overall is difficult for me to understand.**
>
> R1: We will address this weakness in the next version. Please check other responses below too.
>
> **Q2: In certifying the space of attack hyperparameters a model is robust to, it is unclear what hyperparameters are being varied in each attack and what hyperparameter values the work is certifying are safe.**
>
> R2: Fig. 1 and 2 use the default hyperparameters of AutoAttack and SquareAttack. Thus, we only change the attack budget in these two attacks and if the resulting p-value is smaller than 0.05, we reject the null hypothesis, hence we declare the model is safe. Table 1 of page 4 summarizes the hyperparameters we changed in our experiment.
>
>
> **Q3: The experimental results in section 5 seem to suggest the certification is unreliable.**
>
> R3: The randomness comes from the attack. We are running additional experiments where the attack is run multiple times so that we can report more stable p-values
>
> **Q4: Theorem 4 seems to say that multiple rounds of approximation are needed to certify a machine learning model, but it does not quantify or attempt to make a statement about that number of rounds of approximation.**
>
> R4: The supplementary material shows how the number of rounds T needs to scale to retain the ($\alpha$,$\zeta$) model safety guarantee. We agree with the reviewer it is important to state these in the main paper, so we will revise the new version accordingly.
>
> **Q5: What is the relation between this work and PAC (Probably Approximately Correct) learning / adversarial PAC learning? It seems the guarantees are very similar to those proposed in this work.**
>
> R5: (Adversarial) PAC learning estimates the generalization error of a learning algorithm in the presence of an adversary, while our work certifies the adversarial population risk of a pre-trained classifier. Therefore, both approach consider slightly different problems / challenges.
>
> **Q6: Why is this not the case and why is there so much variability in p-values?**
>
> R6: The randomness comes from the attacker, e.g., random sampling squares in SquareAttack. It is possible to reduce the variability by running the attack over additional trials. We will run more trials and report the mean and variance in the next version.
>
> **Q7: Section 2 states "... RS (randomized smoothing) is limited to certifying empirical risk of a machine learning model on pre-defined test datasets under &l_2&-norm bounded adversarial perturbations." What is meant by a pre-defined test dataset and how is randomized smoothing restricted to it? How is PROSAC not restricted to it?**
>
> R7: A pre-defined test set means the RS algorithm only certifies samples in the given test set. Our method in turn offers population risk guarantees.
>
> **Q8: Section 2 states "randomized smoothing (RS) represents a versatile certification methodology free from model architectural constraints or model parameters access" but then later states "Our certification framework shares RS’s versatility but a) it also exhibits the ability to accommodate a diverse range of lp norm-based adversarial perturbations; b) it is not restricted to particular model architectures...". These statements seem to first state that RS is free from model architectural constraints but then states it is restricted to particular model architectures. Do I misunderstand what is being said?**
>
> R8: This is a typo, we will revise this.
>
> **Q9: In Table 1, why is the hyperparameter field "N.A." for AutoAttack? There are several hyperparameters that can be set (e.g., number of gradient steps, number of expecation over transformation estimates, the type of attack to execute, etc.).**
>
> R9: We use the standard version of AutoAttack as most adversarial defense/attack papers do.

---

> ### Author Response · Authors · 2023-11-22
> **Response to Reviewer yeRm - continue**
>
> **Q10: Equation 10 is difficult for me to understand. What is $h_1$? Could a narrative be given for what this equation is saying?**
>
> R10: $h_1$ is defined within the statement of the Theorem. This equation defines how to calculate a super uniform p-value associated with our hypothesis testing problem, allowing us to accept or reject the null, depending on how the p-value compares to $\zeta$.
>
>
> **Q11: In section 4.1, a footnote says that the attack budget and norm are not considered hyper-parameters because it would not be possible to control the risk if the adversary can choose any attack budget. This justification does not explain why the norm is not considered a hyper-parameter. Is there a reason the norm is not considered a hyperparameter?**
>
> R11: In the research community of adversarial machine learning, the norm is often an adversarial attack setting as different norms measure different distances and are not directly comparable. We do not consider the attack budget to be a hyper-parameter because it is expected the attacker will limit the attack budget due to a variety of practical constraints (including limiting the ability of the attack to be detected)
>
> **Q12: In equation 7, shouldn't the risk no longer have a $\lambda$ subscript?**
>
> R12: Yes, we will fix it in the next version.
>
> **Q13: In the paragraph below equation 5, I don't understand what is meant by "We will be assuming in the sequel, where appropriate,..." and later "We will be representing in the sequel..." What is the meaning of the word sequel here?...**
>
> R13: “In the sequel” means “from now on” in English.
>
> **Q14: Is $\mu$ and $\sigma$ the mean and variance of $\lambda$?**
>
> R14: They are mean and variance of GP prediction.

---

### Official Review · Reviewer_z9fX · 2023-11-01

**Soundness:** 2 fair
**Presentation:** 2 fair
**Contribution:** 2 fair
**Rating:** 3
**Confidence:** 2

**Summary:**

This paper derives provable statistical guarantees on the adversarial population risk given an attack algorithm, by computing p-values. This paper also uses a Gaussian Process Upper Confidence Bound (GP-UCB) algorithm for certification against attacks with set of hyperparameter configurations.

**Strengths:**

* This paper derives statistical guarantees on the adversarial *population* risk, which is different from many previous methods for machine learning certification.
* This paper considers that the attack algorithm is known, but it allows the hyperparameters of the attack algorithm to vary within a set of configurations, which is different from previous works on the population risk.
* The proposed method is independent from model architectures, and the experiments applied the proposed method on models including ViT and ResNet.

**Weaknesses:**

* There are lots of existing works on machine learning's robustness certification. Those works have been mentioned in Section 2, but that is probably too late. The first section does not mention the robustness certification works which are not about population risks. It is unclear from the beginning of the paper how this work differs from the previous works, and title is also not sufficiently informative.
* Compared to the existing machine learning certification algorithms that are independent from attack algorithms, the one proposed in this paper requires a specific attack algorithm, which is not applicable when an attacker uses a different attack algorithm but still follows the same threat model. Thus, the importance of such a certification scheme is unclear.
* It is unclear what the computational cost is, and how the number of data samples may affect the results.

**Questions:**

* See the last weakness point.

---

> ### Author Response · Authors · 2023-11-22
> **Response to Reviewer z9fX**
>
> We thank the reviewer for the insightful comments. Here are our responses.
>
> **Q1: There are lots of existing works on machine learning's robustness certification. Those works have been mentioned in Section 2, but that is probably too late.**
>
> R1: We will re-write the title, abstract and introduction to highlight the difference between our work and existing certification methods.
>
> **Q2: Compared to the existing machine learning certification algorithms that are independent from attack algorithms, the one proposed in this paper requires a specific attack algorithm.**
>
> R2: We believe that certifying the population-level dependence on attacks is practical, as users are inclined to seek safety guarantees tailored to their risk exposure for both black-box and white-box attack settings individually. To approximate attack-independent certification, we certify the model – using identical tools – against a wide range of attacks.
>
> **Q3: It is unclear what the computational cost is, and how the number of data samples may affect the results.**
>
> R3: The computational cost derives mainly from the p-value computation in equation 10 , since it involves the computation of the model empirical risk in the presence of the adversarial attack (which also involves optimization of the adversarial perturbation). Note that this computation also involves a maximization over the set of adversarial hyper-parameters over a chosen grid.. With more calibration samples, the variance of the empirical risk appearing in equation would be reduced and the p-value too.

---

### Official Review · Reviewer_sSbB · 2023-11-05

**Soundness:** 1 poor
**Presentation:** 2 fair
**Contribution:** 3 good
**Rating:** 3
**Confidence:** 3

**Summary:**

This submission proposes a new approach to certify the performance of machine learning models against adversarial attacks, in the sense of asserting the model's population risk is lower than some threshold with high probability for a range of hyperparameters of a given attack. Experiments on a few image-based models and both white and black attacks demonstrate the effectiveness of the proposed approach.

**Strengths:**

- The rigorous risk control is a critical problem in trustworthy machine learning, given the legal requirements. The submission tackles this problem with an effective approach.

- The proposed approach is scalable in terms of certifying large ViT models, and experiments cover a wide spectrum of attack methods.

**Weaknesses:**

- The submission may not be rigorous enough. Especially, Theorem 4 only states that "we can do sth by relying on Alg 1". But how is the Algorithm 1's result used to derive the final guarantee in Eqn. (12)? As a certification approach, this process needs to be made more clear. Furthermore, GP-UCB provides maximized $p$ value under some latent assumptions if I understand Appendix D correctly. If this is the case, such assumptions should be inherited in the main theorem under which the certification holds.

- The experimental evaluation is not quite clear and may lack some baselines. For inference, on page 8, the submission has the text "We use $\alpha = 0.10$ and $\zeta = 0.05%$ in the safety certification". However, most results in the paper are presented in terms of $p$ value. How are $p$ values connected with these fixed certification parameters?

- The certification may be a bit limited compared to other $L_p$-norm-based certification, where this work can only guarantee the population risk for a certain type of attack but the existing literature can guarantee the risk for any attack within the perturbation budget. These constraints may need to be made clear.

Minor typos:
1. On Page 3, "relies on ReLU activation"
2. On Page 6, "Fix the machine learning model M, and fix ..."
3. On Page 9, "to the default one in Croce & Hein ..."

**Questions:**

See the questions and suggestions above.

**Details Of Ethics Concerns:**

I may not have the expertise to rigorously evaluate whether the proposed approach can fully align with the requirements of safety and trustworthiness from AI regulations and laws, and such alignment appears to be the main motivation for the proposed method.

---

> ### Author Response · Authors · 2023-11-22
> **Response to Reviewer sSbB**
>
> We thank the reviewer for the insightful comments. Here are our responses.
>
> **Q1: Theorem 4 only states that "we can do sth by relying on Alg 1". But how is the Algorithm 1's result used to derive the final guarantee in Eqn. (12)? As a certification approach**
>
> R1: Theorem 4 uses a regret bound for the GP-UCB algorithm to give a guarantee for the GP-UCB algorithm’s p-value approximation. The reviewer is correct that Theorem 4 uses some assumptions that have been made clear in Appendix D. In particular, we assume that p-value lies in an RKHS with some known kernel k and the observation noise is R-sub-Gaussian. We will make this explicit in the new version in the main Theorem.
>
> **Q2: The experimental evaluation is not quite clear and may lack some baselines. What are $\alpha$ and $\zeta$**?
>
> R2: $\alpha$ represents an (upper bound) to the max adversarial risk, whereas the parameter zeta represents the confidence level (the type-I error probability associated with the underlying hypothesis testing problem).  Thus, we reject the null (ie we declare the model is safe) provided that p-value<&\zeta&. Our figures (Fig. 1 and 2) allow us to establish immediately whether reject the null, hence whether the model is safe.
>
> **Q3: The existing literature can guarantee the risk for any attack within the perturbation budget.**
>
> R3: Yes, our certification is attack-dependent but gives a population-level certification. We will make it clear in the next version. However, we believe that certifying the population-level dependence on attacks is practical, as users are inclined to seek safety guarantees tailored to their risk exposure for both black-box and white-box attack settings individually. To approximate attack-independent certification, we certify the model – using identical tools – against a wide range of attacks.
>
> **Q4: Minor typos.**
>
> R4: Thanks for the comment, we will fix it in our next version.